# RDD-YOLOv5: Road Defect Detection Algorithm with Self-Attention Based on Unmanned Aerial Vehicle Inspection

**DOI:** 10.3390/s23198241

**Published:** 2023-10-03

**Authors:** Yutian Jiang, Haotian Yan, Yiru Zhang, Keqiang Wu, Ruiyuan Liu, Ciyun Lin

**Affiliations:** 1College of Transportation, Jilin University, Changchun 130022, China; jiangyt1720@mails.jlu.edu.cn (Y.J.); yanht1720@mails.jlu.edu.cn (H.Y.); liury1721@mails.jlu.edu.cn (R.L.); 2College of Communication Engineering, Jilin University, Changchun 130022, China; zhangyr1720@mails.jlu.edu.cn (Y.Z.); wukq2020@mails.jlu.edu.cn (K.W.); 3Jilin Engineering Research Center for Intelligent Transportation System, Changchun 130022, China

**Keywords:** deep learning, YOLOv5, road crack detection, transformer, GELU

## Abstract

Road defect detection is a crucial aspect of road maintenance projects, but traditional manual methods are time-consuming, labor-intensive, and lack accuracy. Leveraging deep learning frameworks for object detection offers a promising solution to these challenges. However, the complexity of backgrounds, low resolution, and similarity of cracks make detecting road cracks with high accuracy challenging. To address these issues, a novel road crack detection algorithm, termed Road Defect Detection YOLOv5 (RDD-YOLOv5), was proposed. Firstly, a model was proposed to integrate the transformer structure and explicit vision center to capture the long-distance dependency and aggregate key characteristics. Additionally, the Sigmoid-weighted linear activations in YOLOv5 were replaced with the Gaussian Error Linear Units to enhance the model’s nonlinear fitting capability. To evaluate the algorithm’s performance, a UAV flight platform was constructed, and experimental freebies were provided to boost inspection efficiency. The experimental results demonstrate the effectiveness of RDD-YOLOv5, achieving a mean average precision of 91.48%, surpassing the original YOLOv5 by 2.5%. The proposed model proves its ability to accurately identify road cracks, even under challenging and complex traffic backgrounds. This advancement in road crack detection technology has significant implications for improving road maintenance and safety.

## 1. Introduction

Road safety is a crucial task for transportation [1]. With the increase in the number of vehicles and the continuous growth in road usage, urban roads and highways are experiencing varying degrees of deterioration. The hot summer season exacerbates the situation as the high volume of vehicles generates significant heat, leading to road cracking and collapse. Once the road becomes damaged, the deterioration process accelerates, giving rise to more severe road defects [2]. These road defects not only pose risks to traffic safety but also impact the overall urban landscape, reducing the service life of the roads. Studies by Teschke and Ahlin [3,4] have shown that driving on roads with poor road conditions results in increased vibration, adversely affecting vehicle occupants’ health. Excluding regular traffic facilities such as manhole covers and construction sites, potholes and irregular road cracks should be treated as abnormal features requiring prompt identification [5]. Timely road detection holds immense potential in reducing maintenance costs, prolonging the service life of roadways, and enhancing ride comfort for road users.

The road detection process comprises two main steps. The first step involves inspectors collecting road image information using various carriers. The second step entails employing different detection methods to process and analyze the acquired road images.

In the first step, various methods are employed to capture road defect images. They have evolved from manual and semi-automatic to fully automatic detection. Traditional manual road detection has gradually become obsolete due to its inefficiency and time-consuming nature. Semi-automatic inspection technology refers to the inspection process of storing the collected detection defect images on a hard disk and then manually tracking the cracks by an operator to mark and analyze the cracks. Indeed, both of them have faced the problems of subjectivity and lower accuracy. To address these limitations, an increasing number of studies have turned to using mobile filming equipment for road image acquisition [6,7]. This approach promises to improve efficiency and accuracy in the road detection process. Mei et al. [7] used less expensive devices, such as smartphones, to acquire images of road surfaces and created a dataset containing 600 images of road cracks. In current studies, road inspection vehicles with laser sensors are more accurate [8,9,10,11]; however, they are expensive and difficult to popularize. Nowadays, UAVs, with their small size, low cost, flexibility, mobility, and ability to perform multiple-road parallel inspections, are gaining importance in structural health performance inspections of civil engineering infrastructure [12,13,14,15]. Zhu et al. [16] proposed a dataset with 3151 road defects collected based on UAVs. With its low price, UAVs are gradually becoming a road and bridge detection tool for small and micro-maintenance enterprises and funding-constrained local governments. However, the dataset was collected in a limited location and context, covering only Dongji Avenue in Nanjing, Jiangsu Province, China.

In the second step, road detection algorithms are classified into two categories: traditional computer image processing, including machine learning and computer vision techniques in the framework of deep learning. Traditional computer image processing techniques are currently more mature, such as filter detection methods [17], road defect spectral index [18], local binary pattern methods [19], and machine learning methods like AdaBoost [20]. These methods commonly suffer from limited generalization capabilities and low detection accuracy when applied to diverse road conditions. With the rapid development of deep learning, object detection and segmentation are extensively used in the field of defect detection [21,22]. Singh et al. [23] employed a Mask-RCNN on Road Damage Dataset2020 [24]. This demonstrates that utilizing Mask R-CNN to address this issue is as effective as its applications on common object categories. Road defect detection algorithms based on YOLO (You Only Look Once) or VGG neural network models have better results in several road detection tasks [6,16,25,26]. Wu et al. [27] proposed YOLOV5-LWNet devices for road damage detection on mobile terminals. However, there remains a limited number of studies on combining UAV-based road defect object detection with deep learning. There are several problems: (1) the current UAV-based detection datasets are limited for their number of samples, lacking representativeness and generalization capabilities; (2) existing studies have primarily focused on utilizing existing models [16] or employing computer image processing methods to enhance detection accuracy; (3) General-purpose object detection algorithms may excel on datasets like MS COCO and Pascal VOC, but their performance might be somewhat diminished on road defect images due to the substantial variation in shapes present in these datasets. Nonetheless, there exists a scarcity of research focused on modifying algorithms according to dataset characteristics and image features.

This research aims to address the challenges of high cost and low efficiency in road detection tasks by utilizing unmanned aerial vehicles (UAVs) for road inspections. To overcome the issue of limited dataset universality, we meticulously collected diverse data to facilitate detection in various scenarios. Through an in-depth analysis of the dataset characteristics and image features, an object detection network was devised and is specifically designed for road defect images, resulting in a more efficient and effective road defect detector.

This research collected 2517 images containing road defects as the Road Defect Dataset, covering a diverse range of road scenarios and possessing good universality. In addition, a data augmentation scheme is performed according to the road materials and image features to enhance robustness, including photometric transformation, geometric transformation, and other methods. To obtain a more suitable object detection algorithm for road detection under the UAV perspective, a network based on YOLOv5 for road defect images is designed. Yang et al. [6] argued that crack-like defects in road defects have similar characteristics to image edges in terms of shape and structure. Explicit Vision Center Block (EVC Block) [28] is considered in the improvement strategies, which not only extracts the distinguishing features between crack-type defects and loose defects in edges but also captures long-distance dependencies from large crack-like defects to perform the classification task better. Guo et al. [29] proposed that images where the detection objects are defects often have distracting factors, which require a more flexible and adaptable detector, such as (1) more background interferences; (2) difficulty in identifying objects from the background; (3) large variation in the size of defects in the same class; (4) cluttered defect locations; and (5) objects obscuring each other. The Swin Transformer Block [30] is applied to facilitate the model of high correlations in visual signals, having a more dynamic perceptual field and more flexibly acquiring road defect contextual information. Finally, considering the complex representation of road defect images, Gaussian Error Linear Units (GELUs) [31] are used as the activation function of the model in this study. The experiments demonstrated that the nonlinear capability of the GELU activation function exceeds that of the ReLU [32] or ELU [33]. Furthermore, experimental tricks and UAV flight settings were proposed to boost detection performance and stability in real-world flight scenarios.

The main contributions of this paper can be considered to be as follows:A Road defect dataset is built. It includes a common category of road dataset and covers multiple road backgrounds and traffic conditions with precise manual annotations. To ensure its validity and universality, a few tricks of data augmentations are implemented on the dataset, which is suitable for road defect images based on the UAV perspective.An improved YOLOv5 algorithm named RDD-YOLOv5 (Road Defect Detection YOLOv5) is proposed. Considering the complexity and diversity of road defects, bottleneckCSP(C3) is replaced with a self-attentive mechanism model called SW Block, and a spatial explicit vision center called EVC Block is added to the neck to capture long-range dependencies and aggregate local critical regions. Finally, the activation function is replaced with GELUs to boost its fitting ability.The experiment establishes a UAV flight platform, including an accurate mathematical model for flight altitude and flight speed, to improve UAVs’ efficiency and image quality in collecting road defect images. A few experimental tricks are provided to enhance the performance further. In this case, the anchors are recalculated to improve the positioning accuracy for the abnormally shaped bounding boxes included in the dataset developed in this study. Additionally, label smoothing is applied to mitigate the impact of manual annotation errors in training, enhancing precision.

The remainder of this paper is organized as follows: Section 2 reviews related studies on pavement crack detection. Section 3 provides a detailed description of the structure and features of the dataset, along with the specific data augmentation procedures employed. In Section 4, the network architecture of RDD-YOLOv5 is introduced. Section 5 provides a comprehensive overview of the experimental process, comparison, and ablation experiments. Experiment results explore the performance of the proposed method. Section 6 presents a detailed discussion that delves into the experimental results, analyzing their validity and practical applications. Section 7 summarizes the research findings and potential future work.

## 2. Related Work

To draw a clear distinction between our study and previous research, the literature on vehicle-based road defect detection and UAV-based road defect detection is reviewed. In addition, the literature on image-based defect recognition algorithms is also reviewed.

### 2.1. Detection Carrier

Road defect detection technologies and systems have evolved over the past two decades. Most road detections are based on vehicles and cameras with video imaging technology. Cameras used for detection mainly involve area scan cameras and line scan cameras. Area scan cameras can capture larger areas and specialize in detecting changing objects. D. et al. [34] proposed a method to detect road status based on road images obtained from in-vehicle cameras. Research has shown that applying digital image processing techniques achieves more than 80% processing accuracy. The Commonwealth Scientific and Industrial Research Organization (CSIRO) of Australia has developed a vehicle-mounted automatic road crack detection system called RoadCrack, which is fitted with a line-scan camera and uses digital image pre-processing to identify cracks more prominent than 1 mm in width on highways [35]. The ZOYOM-RTM road detection system was developed by Wuhan Wuda Excellence Technology Co. in China. Detecting crack-type road defects is effective, but the precision of loose-type defects needs improvement [36]. Currently, the road defect detection system based on 3D laser scanning technology can obtain the detailed 3D contours of the road [37]. However, the reliability of this detection method needs to be further enhanced. Although the vehicle-based detection technology can operate under high-speed conditions and achieve high recognition accuracy, it is costly, challenging to popularize on a large scale, and unsuitable for small road maintenance enterprises. In addition, subject to the road inspection vehicle camera view range, engineering personnel often need to make multiple detections of the road, which undoubtedly reduces the efficiency of road inspection.

With the development of science and technology, UAV technology continues to mature. The use of UAVs for inspection offers several advantages: (1) UAVs are deployed flexibly, transmitted, and controlled by remote wireless digital signals; (2) UAVs can capture high-definition images with a wide image perspective and few blind spots. With its unique advantages, it is gradually becoming one of the main carriers of industrial detection. The research topics of UAVs have been widely used in bridge detection, power grid detection, lane line, and vehicle detection [38,39,40], yet their applications in road defect detection are still relatively few. In road defect detection, Jin et al. [41] proposed acquiring images by drone inspection and establishing a highway database to improve detection efficiency. Y.J. Joo et al. [42] argued that current road maintenance inspection vehicles cannot detect difficult-to-access branch roads or side streets. They calculated road areas of interest by using edge-detection techniques for UAC images. However, their dataset only had 470 road images and only collected road conditions at Inha University, which was not representative. Zhu et al. [16] established a flight platform for UAVs and collected road images based on this platform. However, this dataset still has the disadvantage of a single background and is unrepresentative. In addition, this study cut and filled the 300 full-size images collected, obtaining a dataset with 3151 images containing only road defects and losing traffic background noise.

There are still relatively few studies on road defect detection based on UAV inspection and few relevant road defect datasets. Therefore, this research adopts the approach of UAV inspection and takes advantage of it to construct a road defect dataset with multiple backgrounds and disturbances under the UAV perspective.

### 2.2. Detection Method

Detection techniques are divided into traditional computer image processing based on non-deep learning and deep learning object detection algorithms. This paper reviews the conventional work on road defect detection. A deep learning-based surface defect detection method will be discussed to demonstrate the superiority of this approach.

Many researchers have worked on automating crack detection among the traditional computer image processing algorithms. They classified their work into five categories: (1) image thresholding; (2) wavelet transform; (3) edge detection-based methods; (4) classification based on machine learning methods; and (5) minimum path-based methods [6]. Ahmed et al. [43] applied a fuzzy classification method for thresholding, but the false detection rate is high for hard-to-process images. Li [44] proposed threshold segmentation and image enhancement for Bluetooth headset images using the canny edge algorithm for detection. Oliveira et al. [45] established a crack segmentation system. It started by comparing anisotropic diffusion, wavelet, and morphological filtering. However, it has the drawback of complex detection processes and general accuracy, not being suitable for road detection. No matter how well these methods perform in crack detection and segmentation tasks, threshold-based segmentation methods only account for luminance features and are very sensitive to background noise [46]. Wang et al. [47] proposed a wavelet energy field method for pothole detection. It is challenging for governments with tight fiscal expenditures. In short, wavelet transforms or filtering can compensate for the shortcomings caused by noise interference, but it is sometimes difficult to achieve good performance in high-resolution detection tasks. Oliveira et al. [48] used a Bayesian classifier based on a machine-learning approach for crack detection. This approach consists of dividing the road images into several sub-images. As a local method, it is difficult to find complete cracks on the whole image using this method [49]. Amhaz et al. [50] used the minimal path approach to detect road crack lesions and were able to estimate the crack thickness.

The role of deep learning in computer vision is increasingly not to be ignored. Detection methods using CNNs to extract abstract and representative features have achieved outstanding results in localizing and classifying surface defect detection tasks [51]. Deep learning is free from the interference and limitation of the object’s background and works well in multiple detection scenarios. Object detection and instance segmentation focus on recognizing objects within images. The former identifies the position and category of objects, while the latter goes a step further by categorizing each pixel for each of them. In road defect detection using unmanned aerial vehicles (UAVs), the emphasis is on the precise identification and localization of objects (defects) on the road. Therefore, this study delves into developing and reviewing object detection techniques to explore how these technologies can be applied to achieve accurate recognition and localization of road defects in UAV-based detection. Further, object detection algorithms do not rely on high-precision and high-resolution cameras, depending only on the gimbal camera carried by the UAV to complete road defect detection, which greatly reduces the cost of road detection. Deep learning-based object detection algorithms are divided into two main categories: single- and two-stage object detection algorithms. Two-stage object detection algorithms usually involve two processes of detection: firstly, generating a set of sparse regional proposals; secondly, classifying and regressing the regional proposals [52]. This method (1) handles the imbalance of categories and (2) uses two-stage features to describe the object, leading the two-stage approach to a higher detection accuracy. When comparing the mainstream two-stage object detection algorithms among R-CNN [53], Fast R-CNN [54], and Faster R-CNN [55], it is worth noting that they all share a common feature: they use a VGG network as the backbone network, demonstrating the robustness. However, the inference speed of these methods is slower than single-stage object detection algorithms, endearing them of limited practical significance. The single-stage object detection algorithm directly generates the object’s category probability and position coordinate values by sampling the position, scale, and aspect ratio without developing regional proposals. This method achieves object detection by extracting features only once and is more computationally efficient. There are mainstream algorithms such as YOLO series and SSD (Single Stage Detection). In practical industrial detection, the single-stage detection approach is gaining importance considering the carrying capacity of the equipment and equipment deployment. Zhu et al. [16] compared three typical object detection algorithms: Faster R-CNN, YOLOv3, YOLOv4, and improved-YOLOv3 [56] and YOLOv4 variants to select the most suitable road defect images for UAV view. Vision transformer shows stronger robustness in object occlusion datasets [57] but generates high computational complexity. To solve the problem of many disturbing elements and serious object occlusion in UAV-captured images, Zhu et al. [58] proposed TPH-YOLOv5. They replaced the previous prediction heads with Transformer Prediction Heads and added a Convolutional Block Attention Module (CBAM) [59] to process the aerial images. However, the transformer structure generates very high computational complexity on the image. Although TPH-YOLOv5 significantly increased the detection accuracy of small objects under UAV aerial photography, it raised high requirements for industrial deployment and is not suitable for lightweight road defect detection.

Few studies and relevant road defect datasets are related to road defect detection based on UAV inspection. Therefore, this research adopts UAVs as detection carriers and takes advantage of it to construct a road defect dataset with multiple backgrounds and disturbances under the UAV perspective.

## 3. Dataset

Existing road defect datasets suffer from limitations such as scenario homogeneity and sample insufficiency. For instance, the public dataset CRACK500 [54] covers only 500 images with pixel-level annotated binary maps. Zhu [16] collected a dataset containing only 300 full-size images from a single road, which were subsequently processed and segmented to obtain 3191 images without other backgrounds and transportation facilities. To address these limitations, a comprehensive collection of representative road defect images is conducted from multiple regions across China. The dataset comprises 2517 high-resolution images of different sizes with diverse scenes.

This section introduces detailed information about the dataset’s content and statistical characteristics. Additionally, a data augmentation scheme is proposed to overcome the statistical limitations and improve the dataset’s quality. The Road Defect Dataset is open access and can be found at https://github.com/keaidesansan/Roadcrack_Dataset_2517.git. (accessed on 26 September 2023).

### 3.1. Dataset Collection

The Road Defect Dataset covered multiple provinces across China, including Hebei Province, Jilin Province, Henan Province, and Shandong Province. In order to increase the universality of training results in the context of road defects, it is beneficial to collect corresponding images from various road scenarios, helping the model to understand the variations and patterns of road defects that may occur in natural environments. The dataset contains six scenarios, as shown in Figure 1: (1) country roads; (2) campus roads; (3) urban feeder roads; (4) urban arterial roads; (5) sub-urban roads; and (6) urban expressways. Images were captured at different moments throughout the day to emulate diverse lighting conditions. It can be observed in Figure 1 that road surfaces undergo various degrees of damage through long-term use. The images encompass a comprehensive range of road types, which contributes to the model’s increased transferability.

In order to align with the actual distribution of road defect occurrences and morphological characteristics, this paper categorizes road defects into four main classes: transverse cracks, longitudinal cracks, alligator cracks, and potholes.

### 3.2. Dataset Characteristic

After conducting statistical analysis on the Road Defect Dataset, it contained 2517 images with a total of 14,398 instances. This indicates a substantial volume worth summarizing regularly in terms of its magnitude. The dataset has the following characteristics:

(1) Realistic distribution of defects: Figure 2a shows that the number of different defect classes varies greatly, with longitudinal cracks being the most prevalent. This distribution of defect numbers closely aligns with real-world road damage scenarios, lending authenticity to the findings.

(2) Diverse defect shapes: As shown in Figure 2b, comparing the aspect ratios of bounding boxes from multiple datasets, including the Road Defect Dataset, CrackTree200, CrackForest Dataset, and China Drone from RDD2022, it is evident that the Road Defect Dataset constructed in this study exhibits a relatively uniform distribution of aspect ratios ranging from 0 to 0.1 to 10 and beyond. This indicates the diverse range of shapes of objects in this dataset, presenting a significant challenge for models in terms of calculating target bounding boxes.

(3) Complexity of image features: As shown in Figure 2c, over 80% of the images within the Road Defect Dataset exhibit the presence of two or more instances, thereby offering more intricate image features compared to the Pascal VOC dataset, showing more similarities to MS COCO. The complex nature of these images necessitates the development of robust algorithms capable of effectively detecting and distinguishing multiple instances within a single image.

(4) Diverse instance sizes: As shown in Figure 2d, in CrackTree200, CrackForest Dataset, and China Drone from RDD2020, the majority of images contain only a single instance, which places relatively lower demands on the model’s performance. Instead, the majority of road defects within the dataset exhibit varying sizes, typically ranging from 0.2% to 16% of the image size when observed at high resolution. Statistical analysis determined that the largest road defects account for approximately 69% of the total area, further emphasizing the diversity in instance sizes. Consequently, the model faces a substantial challenge in effectively extracting robust features at different scales to accurately identify and analyze road defects.

Summarizing the aforementioned features, the Road Defect Dataset is a comprehensive and dependable resource comparable to datasets of similar nature. However, these characteristics also present three notable challenges: (1) The uneven distribution of defect categories; (2) the precision of road defect identification; and (3) the accuracy of defect localization. The upcoming work aims to devise strategies to address these challenges.

### 3.3. Dataset Augmentation

To address the first challenge presented in 3.1, the uneven distribution of defect categories, this paper proposes a data augmentation strategy. Unlike simplistic approaches such as mirroring or adding noise, data augmentation of this research was categorized into three schemes: photometric distortion, geometric distortion [60], and intelligent transformation. These schemes effectively expand the dataset by simulating various shooting scenes. Photometric transformations involve adjustments to brightness, hue, and other parameters, enhancing the dataset’s diversity. It is critical to exercise caution during photometric transformations to avoid wearing out the crucial details of road defects. Geometric transformations encompass changes in image orientation and shape. However, it is essential to avoid unsuitable geometric transformations. For instance, crack-like defects heavily depend on features themselves, such as the direction of extension. Excessive geometric transformations may invert the annotations, leading to misclassified images. Intelligent transformations, primarily implemented through Mix class algorithms, play a crucial role in sample enhancement by facilitating feature migration, including MixUp [61], CutMix [62], and FMix [63]. MixUp has the ability to generate new training samples by linearly interpolating between multiple samples and their labels. CutMix is a variation of MixUp that randomly cuts out a rectangular region of the input image and replaces it with a patch from a different image in the training set. FMix is another variation of CutMix that cuts out an arbitrarily shaped region and replaces it with a patch from another image.

In summary, this paper proposes a set of 14 data augmentation methods that serve as a data augmentation strategy of “bags of tricks”, filtering out ineffective techniques in order to enhance the model’s robustness for road defect images. Photometric transformation contains Gaussian blur, Gaussian noise, Poisson noise, brightness adjustment, and hue adjustment. The photometric transformation includes random crop, random shift, random rotation, random flipping, mirroring, and random cut-out. Mix transformation has MixUp, CutMix, and FMix. Notably, an image’s brightness, contrast, and blur will affect its accuracy. For the same image, the road defects will be severely distorted when the photometric transformation scale factor exceeds a certain range. Increasing the false detection rate, i.e., background FP, is possible. Thus, the photometric transformation employed in our methodology incorporates the crucial determination of the scale factor. The process of factors are as follows:Gaussian blur

Gaussian blur smooths the image and simulates the motion blur effect caused by flying drone shots. This effect simulates the inherent motion and vibrations encountered during image capture, mitigating unwanted jaggedness or pixelation. Sometimes, Gaussian blur reduces the disturbances caused by Gaussian noise. An effective range of parameters is available to simulate the blurring of images captured by the UAV while in flight.
(1)$$I_{1}^{\prime }(x,y)=(K⊗I)(x,y);$$
where $I_{1}^{\prime }(x,y)$ is an image after Gaussian blur is applied. K is the Gaussian kernel. I is the original image. ⊗ is the 2D convolution operation.

2.Gaussian Noise

Gaussian noise stands as a prevalent technique employed to bolster the model’s robustness in deep learning. It is noteworthy that excessive noise may result in the loss of image detail and quality. So, it is important to find appropriate parameters, namely the mean and variance of the Gaussian noise.
(2)$$I_{2}^{\prime }(x,y)=\mu +\sigma \cdot Z(x,y);$$
where $I_{2}^{\prime }(x,y)$ represents images after Gaussian noise. Z(x,y)~N(0,1).

3.Poisson noise

Poisson distribution serves as a powerful tool to model the number of photons detected by each pixel in an image, simulating statistical characteristics of photon noise under low-light conditions. Incorporating Poisson distribution into the road defect detection task will simulate outdoor lighting and weather changes.
(3)$$I_{3}^{\prime }(x,y)=I(x,y)+Poisson(I(x,y)\cdot p_{scale});$$
where $I_{3}^{\prime }(x,y)$ represents images after Poisson noise. Poisson represents the Poisson distribution. Notably, $p_{scale}$ should be added to reduce the effect of photonic noise, representing the parameter scaling factor for the Poisson distribution.

4.Brightness adjustment

Adjusting the brightness of an image simulates a range of lighting conditions that are encountered during testing, thereby enhancing the model’s adaptivity to variations in lighting. Effective brightness adjustment techniques require limiting the appropriate lighting parameters.
(4)$$I_{4}^{\prime }(x,y)=\alpha \cdot f(x,y)+\beta ;$$
where $I_{4}^{\prime }(x,y)$ represents images after adjusting brightness. α is a scaling factor that adjusts the overall brightness of the image. β is an offset value that controls the brightness level of the image.

5.hue adjustment

Different road materials exhibit distinct colors. Among the commonly used road materials, asphalt and concrete stand out prominently. In the HSV color space, the hue value of an asphalt road is approximately 30–40 (yellow-orange), while that of a concrete road is around 0–10 (gray). Slight variations in hue values occur under different lighting conditions, adding to the natural complexity of the environment. In the research, hue adjustment is confined to 180–230 to ensure consistency.
(5)$$I_{5}^{\prime }(x,y)=[H(x,y)+H^{\prime }]\mathrm{mod}180;$$
where $I_{5}^{\prime }(x,y)$ represents images after adjusting hue. H is the original hue value of a pixel. $H^{\prime }$ is the hue shift value.

## 4. Methods

This section presents a concise overview of the architecture of RDD-YOLOv5, while emphasizing the location, mechanism, and advantages of three key modules: SW Block (Swin Transformer Block [30]), EVC Block (Explicit Vision Center Block [28]) and CBG (Convolution Batch Normalization GeLU [31]), which are integrated into RDD-YOLOv5 to enhance its performance. In summary, the SW Block elevates the model’s capacity to capture long-range dependencies, the EVC Block improves the extraction of road defect features, and the replacement of GELU activation contributes to the overall performance enhancement.

### 4.1. Introduction of YOLOv5

YOLO series algorithms divide the image into equally sized grid cells and infer class probabilities and bounding boxes for each of them based on a convolution neural network (CNN), including four parts: input, backbone, neck, and prediction head.

Input: YOLOv5 adaptively computes the initial anchor sizes. During the training stage, the model predicts the bounding boxes based on the anchors and then compares them to the ground truth to calculate the discrepancies. Subsequently, the network parameters are updated iteratively to minimize the differences in the backward propagation. Additionally, multiple techniques are incorporated in the YOLOv5 to enhance its inference speed, such as Mosaic data augmentation and adaptive image scaling.

Backbone: The focus module performs a slice operation on the input. Compared to the conventional downsampling method, it reduced information loss. In YOLOv5-6.0, the Focus module has been replaced with convolutional layers (Conv) with kernel size 6, stride 2, and padding 2. Based on the CSP (Cross Stage Partial) architecture, the C3 module is the primary module for learning residual features. It consists of two branches. One branch utilizes multiple stacked Bottleneck blocks and three standard convolutional layers, while the other branch passes through a basic convolutional module. Finally, the outputs of both branches are concatenated. SPPF takes SPP’s place in YOLOv5 for its faster performance. SPPF takes the input and passes it through multiple MaxPool layers with different sizes in parallel and then further fuses these outputs. This approach helps address the multi-scale nature of the detection task.

Neck: A combination of the Feature Pyramid Network (FPN) and the Path Aggregation Network (PAN) structures is employed in YOLOv5. FPN constructs the classical structure of a feature pyramid using a top-down side-by-side connection that builds a high-level semantic feature map at all scales. After multiple network layers, the target information at the bottom is already very ambiguous. PAN structure incorporates bottom-up routes to compensate for and strengthen the localization information.

Prediction head: YOLOv5 produces predictions at three scales: small, medium, and large size. The model calculates total loss consisting of three parts: classification loss (BCE Loss), confidence loss (BCE Loss), and bounding box loss (CIOU Loss). Notably, the confidence loss is assigned weights on the three prediction scales. These processes are shown in Equations (6).
(6)$$\begin{array}{l} Loss=\lambda_{1}L_{\mathrm{cls}}+\lambda_{2}L_{\mathrm{obj}}+\lambda_{3}L_{\mathrm{loc}};\\ L_{\mathrm{obj}}=4.0\cdot L_{\mathrm{obj}}^{\mathrm{small}}+1.0\cdot L_{\mathrm{obj}}^{\mathrm{medium}}+0.4\cdot L_{\mathrm{obj}}^{\mathrm{large}}; \end{array}$$

### 4.2. Overview of RDD-YOLOv5

This work centers around optimizing the backbone and neck to adapt to the requirements of road defect detection. (1) Backbone: Utilizing the property that SwinTransformer Block implements a shift window-based self-attention mechanism on image patches, SW Block is introduced to alter the generation of the smallest-scale feature map owing to the CNN-based YOLOv5. This strategy focuses on the model’s ability to capture long-range dependencies within images. (2) Neck: the EVC Block is embedded in this part. This strategy takes advantage of the multi-scale feature fusion in the FPN+PAN structure to extract more complex features and fuse them with shallow-layer features to capture and represent complex information across different scales. (3) Whole structure: YOLOv5-6.0 applied the SiLU (Sigmoid-Weighted Linear Unit [64]) as the activation. The research employed GELUs (Gaussian Error Linear Units) to promote the model’s nonlinear fitting capabilities, albeit at the cost of slightly sacrificing convergence speed, leading to a boost in the overall performance. The structure of RDD-YOLOv5 is shown in Figure 3. The arguments of the network is shown in Table 1.

### 4.3. SW Block (Swin Transformer)

Figure 2c shows that the largest instance in the dataset has a ratio of about 0.70 to the image size, which is typically represented by alligator cracks. The interference caused by the road surface background can significantly alter the semantic information of road defects, leading to misclassification and mis-segmentation. To enhance the discriminative capabilities for surface defects with larger scopes or complex backgrounds, it is necessary to collect contextual information from a broader neighborhood. However, traditional convolutional networks face limitations in capturing global contextual information due to their focus on local features, which hamper their ability to effectively learn global relationships and model the relationship object-to-object. In computer vision, the interested relationships include pixel-to-pixel, pixel-to-object, and object-to-object. While the first two types of relationships can be modeled using convolutional layers and region of interest alignment (RoIAlign) [65], the modeling of the last type lacks a well-established method.

Convolution Neural Networks (CNNs), primarily composed of convolution and pooling layers, extract features through localized perception of image matrices. However, it solely captures local information, leading to the loss of inter-data relationships. Dosovitskiy et al. [66] proposed Vision Transformer (ViT), a model that utilizes a self-attention mechanism to model object-to-object relationships and exhibits global solid modeling capabilities. Concurrently, ViT poses a challenge for practical applications because of its prohibitively high computational complexity. Liu et al. [30] proposed the Swin Transformer, which reduces the computational complexity of self-attentive computation through window partitioning and shift window operations. The shift window operation allows for information exchange between non-overlapping windows, facilitating communication between feature maps. The self-attentiveness of the window enables a genuinely global focus on dependencies between image feature blocks through window interaction and overcomes computational challenges while capturing long-range relationships. Therefore, Swin Transformer Block (SW) is introduced as an improvement strategy, leading to improved performance in defect classification tasks.

SW Block is an abbreviation for Swin Transformer Block. Its architecture includes two blocks with nearly the same structure as the Swin Transformer proposed by Liu [30], as shown in Figure 4. The first part uses the standard window-based multi-head self-attention mechanism. The second part replaces the standard multi-head self-attention mechanism with shifted window multi-head self-attention. The SW Block is added after the feature map output of $\frac{1}{32}$ size, where the feature map at the end of the backbone contains more semantic information with a smaller image size, decreasing computational complexity. The self-attention mechanism enables effective modeling of the context. The provided Equation (7) offers a means to estimate the computational complexity of the RDD-YOLOv5 model.
(7)$$\begin{array}{l} \Omega (MSA)=4hwC^{2}+2{\left(hw\right)}^{2}C,\\ \Omega (W-MSA)=4hwC^{2}+2M^{2}hwC; \end{array}$$
where Ω(MSA) is the computational complexity of multi-head self-attention modules based on windows. Ω(W−MSA) is the computational complexity of multi-head self-attention modules based on shifted windows. The h and w represents the size of the feature map. C is the channel of the feature map. M is fixed. In this paper, M is set to 7 by default. By performing the shift windows operation, the computational complexity of shifted window-based self-attention increases linearly instead of nonlinearly with the size of the feature map, reducing the model’s computational complexity.

The incorporation of the SW Block enhances its ability to center around object-to-object relationships, enabling the capture of long-range dependencies more effectively. The ablation experiment in Section 5.5 validates the rationale behind the incorporation of the SW Block.

### 4.4. EVC Block (Explicit Vision Center Block)

There are some issues to be solved, such as erroneous image segmentation that leads to the loss of important features, especially for those with extensive areas. For instance, the elongation of a crack may result in part of the feature being unrecognized or wrongly assigned to a different bounding box, leading to multiple defects being labeled on the same image, which will affect the representation of damage degree to the road surface and ultimately impact highway maintenance decisions. Furthermore, identifying various types of road defects relies on recognizing local key features such as curvature. Consequently, better capturing global and local features is important to resolve this problem effectively.

The YOLOv5 neck adopts an FPN + PAN structure, which enables the lower-level feature maps to contain stronger semantic information. This improves the model’s ability to detect features at different scales through upsampling and feature fusion. Quan et al. [28] proposed a Centralized Feature Pyramid network with a spatial explicit visual center (EVC) scheme. EVC Block contains a lightweight MLP and a learnable visual center (LVC). The lightweight MLP captures global long-range dependencies, and the LVC aggregates local key regions. Longitudinal cracks and potholes often have similar backgrounds, which can be resolved by allowing the lower-level feature maps containing positional and detail information to obtain stronger semantic features. Therefore, we introduce EVC Block into the neck network. The addition of EVC Block not only solves the problem of incorrectly labeled large-scale road defects but also further identifies the subtle features of road defects, such as curvature, in deep features.

EVC Block is an abbreviation for Explicit Visual Center Block. It consists of two processes, namely lightweight MLP and learnable visual center (LVC), as shown in Figure 5.

In lightweight MLP, input features $X_{in}$ are normalized by group normalization two times and then passed through a depthwise convolution-based module in the first block and an MLP-based module in the second block. Output features $X_{in}^{i}$ pass through channel scaling and the droppath. The residual connections are implemented. This process can be summarized by the following Equation (8):(8)$$\begin{array}{l} X_{in1}=\mathrm{DConv}(\mathrm{GN}(X_{in}))+X_{in},\\ X_{out1}=\mathrm{CMLP}(\mathrm{GN}(X_{in1}))+X_{in1}; \end{array}$$
where CMLP(•) represents the channel MLP. $X_{in}$ represents the input feature. $X_{in1}$ represents the output of the first block. $X_{out1}$ is the output of the second and the last block.

LVC consists of multiple convolutional layers and encoding operations. After passing through four convolutional layers, the features are inputted into a codebook. Here, the authors used a set of scaling factors to ensure that the features of the input and the factors in the codebook correspond to their respective position information. After the entire image is mapped to the codeword, the outputs are fed into a layer with ReLU and batch normalization. Information output from the previous layer is then fed into a fully connected layer and a convolutional layer. This process is described in Equation (9):(9)$$X_{out2}=X⊕(X_{in}⊗O);$$
where O represents the output from the last convolution layer. $X_{out2}$ represents the output from the LVC block.

Combining the outputs of lightweight MLP and LVC, deeper features will be obtained in Equation (10):(10)$$X_{out}=\mathrm{cat}(MLP(X_{in}),LVC(X_{in}));$$

In RDD-YOLOv5, the EVC Block is added within the neck. After upsampling to generate a larger depth feature map, the EVC Block operation is conducted. Subsequently, the deep features are fused with shallow features, enabling the latter to benefit from semantic information. This process is repeated twice to ensure a comprehensive fusion of shallow features and deep features.

### 4.5. Convolution Batch Normalization GeLU (CBG)

The essence of an activation function is to add non-linearity to the neural network by applying a nonlinear transformation to the linear transformation $w^{T}+b$. In addition to non-linearity, an important property of a model is its ability to generalize, which requires the addition of stochastic regularization. Therefore, the input to a model is determined by both the nonlinear activation and stochastic regularization. Hendrycks et al. [31] proposed the GELU function, which has better nonlinear fitting than ReLUs or ELUs. GELUs stands for Gaussian Error Linear Units, a stochastic regularization equation combining properties from dropout, zone-out, and ReLUs. The authors introduced the idea of stochastic regularization into the activation function, which intuitively fits our understanding of natural phenomena. The approximate calculation for GELUs is Equation (11). As seen, the GELU activation is more complex than others.
(11)$$0.5x(1+\mathrm{tahn}[\sqrt{2/\pi }(x+0.044715x^{3})]);$$

YOLOv5 uses the SiLU as the activation function. As shown in Figure 6a, the SiLU has a smooth and continuously differentiable shape, which helps reduce the possibility of gradient vanishing. The GELU has a shape similar to the SiLU in terms of smoothness and continuity. In contrast to SiLU activation, the GELU is advantageous for capturing more intricate relationships within the data of its slight S-curve. Figure 6b shows that both the GELU and SiLU have non-zero gradients for all input values, but the derivative of the GELU is more complex, which can help the model capture more complex and subtle patterns in the data. In this study, the SiLU is replaced with GELU, and mAP and other metrics are compared in the experiment of Section 5. The experimental results show that replacing the activation function of YOLOv5 with the GELU is a wise choice for the Road Defect Dataset.

## 5. Experiments

This section will introduce flight setup for efficient UAV inspection and other experimental tricks. The UAV flight tuning process includes flight speed and flight altitude. The proposed work’s operating environment includes training and testing on cloud servers. Other training settings and evaluations will be described in the following content.

### 5.1. Flight Setup

Establishing a suitable flight setting for UAV inspection is necessary to operate more efficiently, satisfy data requirements, and ensure the pavements being captured are covered correctly under the UAV camera. When efficiently and effectively capturing road defect images using UAVs, (1) images should preferably contain more than one road to take advantage of the parallel detection of drones; (2) images can realistically simulate the roads under normal traffic conditions and noise; (3) image resolution is guaranteed, i.e., the pavement defects in the images can be discerned by the human eye through the images; (4) UAVs should be far away from road distractions; and (5) images under continuous shooting settings can ensure continuity and independence.

#### 5.1.1. Flight Altitude

An optimal flight height enables the UAV to capture the entire road within its field of view, along with a portion of the surrounding traffic environment, which helps mitigate the potential degradation of image clarity caused by excessively high altitudes. Figure 7a describes the relationship between the drone’s flight altitude and the camera. The actual light rays converge and are presented to the camera. Light rays form a bunch of isometric triangles. Flight altitude should satisfy Equation (12) or Equation (14):(12)$$H=\frac{f\cdot W}{s_{w}};$$
where H represents flight altitude. f represents the focal length of the camera. W represents the width of the road to be inspected. W is influenced by several factors, such as road grade, road width, number of lanes, width of dividers, and green belts. $s_{w}$ represents the camera sensor size. $s_{w}$ can be expressed by Equation (13), as follows:(13)$$s=\sqrt{s_{w}{}^{2}+s_{h}{}^{2}};$$
where s is the diagonal size of the camera sensor. $s_{h}$ is the height dimension of the camera sensor. $s_{w}$ is the width dimension of the camera sensor.
(14)$$H=\frac{W/2}{\tan\frac{FOV}{2}};$$
where FOV is the field of view, which is a fixed parameter. It can be expressed by Equation (15):(15)FOV=2arctan(s/2f);

In addition, it is crucial to factor in flight clearance when determining the appropriate flight altitude for the UAV [16]. Given that roads serve as vital transportation infrastructures, they are often accompanied by various obstructions, including traffic signs, traffic lights, street lights, trees along the roadside, utility poles, and other public amenities. Hence, the UAV’s flight altitude must be adhered to the clearance requirements. Table 2 provides the altitudes commonly associated with typical infrastructural elements.

#### 5.1.2. Flight Speed

Based on the established flight height, an appropriate flight speed effectively reduces image overlap, consequently enhancing the efficiency of UAV road detection tasks. UAV flight speed affects the overlap degree and the clarity of the captured images [16]. The higher the drone speed, the smaller the overlap and the similarity of adjacent images. However, an unsuitable speed causes a motion blur effect in camera shake [67], resulting in obvious artifacts in the captured images, which will blur the road defect lesions in the images. Flight speed is determined based on requirements for overlap and images according to the mathematical equation derived from the road width. The process is shown in Figure 7b.

GSD (Ground Sampling Distance) refers to the distance between ground pixels on the ground in an image, officially provided by DJI, applicable to a wide range of focal lengths. The most prevalent GSD value, commonly seen with a 24 mm focal length, is calculated as H/55. Other GSD parameters are shown in Table 3. By utilizing these GSD parameters, it becomes possible to accurately calculate the real-world length of the road captured by the UAV as Equation (16):(16)$$L=\frac{W}{GSD};$$
where L represents the actual length of the road from drone-captured images. GSD represents Ground Sampling Distance.

In the experiment, UAVs are operated at the same speed and uniform linear motion. The overlap of captured images is r≤50% to ensure the independence of the data. The above process is described by Equation (17):(17)$$\{\begin{array}{c} L_{overlap}=\frac{L}{2}+\frac{L}{2}-vt\\ L_{overlap}\leq 0.5L \end{array};$$
where $L_{overlap}$ represents the overlapping distance of two adjacent images. v represents flight speed. t represents the shooting interval of two adjacent images.

By associating Equations (16) and (17), flight speed with a 24 mm focal length lens can be calculated as Equation (18):(18)$$v\geq \frac{W\cdot H}{110t};$$

Take DJI Mavic3 as an example, using a Hasselblad camera to capture road images. In Section 3.1, Data Collection, the dataset consists of six scenarios that were categorized based on the width of the road and the surrounding infrastructure construction into high-grade roads and low-grade roads, which helps determine the appropriate UAV flight height and speed for each scenario. According to the equations in Section 5.1.1 and Section 5.1.2, the flight altitude and flight speed on the two categories of the road can be calculated. The UAV flight parameter table is shown in Table 4.

### 5.2. Operation Environment

The experiments were trained on a remote server based on the AutoDL platform (NVIDIA RTX 3090 GPU, 24 GB; Intel(R) Xeon(R) Platinum 8350C CPU @ 2.60 GHz 56 GB memory). The YOLOv5 framework was built under the Ubuntu system to implement training and inference for pavement defect target detection.

### 5.3. Experimental Tricks

In the final experiment, two additional optimization techniques are applied to further enhance the architecture: K-Means++ for generating more accurate anchor boxes and label smoothing to reduce loss.

#### 5.3.1. K-Means++

YOLOv5 computed the anchor boxes by applying K-Means, which initializes K clusters being measured the distance from each other sample point. Samples were divided into the clusters with the closest distance. Each sample is partitioned to the cluster centroid closest to it, thus forming a cluster. It remains a problem that the convergence situation is heavily dependent on the initialization of cluster centers, resulting in a risk of falling into local optimization. Moreover, original anchor boxes are practical for MS COCO instead of the Road Defect Dataset, where MS COCO contains mostly large or medium objects, but the majority of the size of objects in the Road Defect Dataset are medium and small, leading the YOLO prediction head to screen out improper bounding boxes. As shown in Figure 2b, the label aspect in the Road Defect Dataset presents more differences from other datasets, requiring the algorithm to reaggregate anchors of different sizes. Means++ was employed to recalculate bounding boxes for a significant number of defects, optimizing the first stage to avoid randomly selected cluster centroids’ just convergences to a locally optimal solution, subsequently reducing the error of the classification results. This process can be described in Equation (19):(19)$$P=\frac{D{\left(x\right)}^{2}}{\sum_{x\in \chi }D{\left(x\right)}^{2}};$$
where D(x) represents the Euclidean distance between initial clustering centroids and samples. Equation (19) calculated the probability that each sample is selected as the next cluster center point, following the roulette wheel method to select the next cluster center until the selection gets K cluster centroids. Keep updating the cluster centroids until the positions no longer change.

Three scales of anchor sizes were evolved through the K-Means++ algorithm. The results are shown in Table 5.

#### 5.3.2. Label Smoothing

Label smoothing is a regularization strategy that reduces the weight of the real sample label categories in calculating the loss function, ultimately suppressing overfitting. A hyperparameter ε, the smoothing parameter of label smoothing, can be adjusted to enhance the model’s robustness. Modifying this parameter makes it possible to steer the model toward a more resilient state. When the classification result is accurate, the probability p=1 is corrected after label smoothing as Equation (20). Instead, the probability p=0 is calculated by Equation (21):(20)$$p_{true}=1-0.5*\varepsilon$$
(21)$$p_{false}=0.5*\varepsilon ;$$

The experiment modifies the label smoothing hyperparameter as ε=0.1 in the final experiment.

### 5.4. Evaluation Metrics

To evaluate the effectiveness of RDD-YOLOv5 in detecting road defects based on the UAV dataset, we adopt P (precision), R (recall), $F_{1}$ (F1 score), and mAP (mean average precision) as network evaluation metrics.

The precision equation is shown in Equation (22):(22)$$P=\frac{TP}{TP+FP};$$
where P represents the precision. TP represents true positives, which are the detection boxes with an IoU (Intersection over Union) greater than the set threshold with the ground truth boxes, being considered correctly identified by the model; FP represents false positives, which are the detection boxes with an IoU less than the set threshold, being considered the object of detection errors.

The recall equation is shown in Equation (23):(23)$$R=\frac{TP}{TP+FN};$$
where R represents the recall rate. FN represents the number of objects that were not detected as positive samples by the model.

F1 score (F1) is the harmonic mean of precision and recall, providing a comprehensive performance evaluation metric for the model and its maximum performance. The equation is shown in Equation (24):(24)$$F1=\frac{2\cdot P\cdot R}{P+R};$$

It is necessary to introduce a parameter that can measure both precision and recall to maintain both precision and recall at a high level simultaneously. AP (average precision) is a metric for evaluating the performance of object detection models. It measures how well a model detects objects of a specific class by integrating precision and recall over different levels of confidence scores and IoU thresholds. AP is computed by calculating the area under the precision–recall curve. A higher AP indicates better performance of the model in detecting objects of a particular class. AP for different classes is calculated by Equation (25):(25)$$AP_{i}=\int_{0}^{1}P(R)d(R);$$

Mean average precision (mAP) is a widely used metric in object detection tasks, which is an important subfield of computer vision. It is used to evaluate the performance of object detection models, which aim to locate and classify objects within an image or video. mAP takes into account both the accuracy and completeness of the model’s predictions, making it a comprehensive metric for evaluating the overall performance of a model.
(26)$$mAP=\frac{\sum AP}{N}$$
where N is the number of object detection classes. There are four classes in the dataset, hence N = 4. The metric mAP@0.5 represents the mean average precision (IoU = 0.5), which calculates the AP of all images in each class at the IoU threshold of 0.5. The metric mAP@.5:.95 represents the average mAP at different IoU thresholds from 0.5 to 0.95 with a step of 0.05. The prominent disparity between the two metrics is that the former metric places importance on the recognition of road defects themselves, while the latter metric emphasizes the precise localization of these road defects.

### 5.5. Ablation Experiments

The purpose of conducting ablation experiments is to validate the effectiveness of the model’s improvement strategies. In this study, the ablation experiments were designed to validate not only the effectiveness of the three proposed modules in the model but also to assess the usefulness of anchors recalculated using the K-Means++ algorithm, given the uniqueness of the constructed dataset and the irregular shapes of the defects. Therefore, this section of ablation experiments includes four modules: SW Block, EVC Module, GELU, and the K-Means++ algorithm for anchor calculation. Each of these four modules was separately integrated into YOLOv5 to make evaluations. The experimental results are shown in Table 6, where “✓” and “🗴” denote the absence and presence of the corresponding strategy, respectively.

As the most lightweight model, YOLOv5 achieved a satisfactory and comprehensive performance. The clever utilization of the clustering trick and three model improvement strategies achieved a notable enhancement in mAP@0.5 and mAP@.5:.95. mAP@0.5 (mean average precision) stands as the most widely utilized metric. mAP@0.5 experiences a subtle increase when employing SW Block and CBG, while the EVC Module results in a more noticeable elevation of about 2%. The cluster trick K-Means++ led to a remarkable improvement with the least amount of effort. Upon the application of all these techniques to the network, mAP@0.5 witnesses a substantial increase of approximately 2.5%. The experimental results indicate that the combination of K-Means++, SW Block, EVC Block, and CBG exhibits a complementary nature, and their individual contributions to network performance have a cumulative effect. Notably, these strategies exhibit a more pronounced enhancement in mAP@.5:.95 that places a high emphasis on the accuracy of object positioning. mAP@.5:.95 displays a more significant improvement when applying the EVC Module, and it makes more contributions to this metric. The convergence visualization is shown in Figure 8.

In addition, to facilitate a more comprehensive comparison, Figure 9 illustrates the precision, recall, precision–recall, and F1 score curves for the four models at varying confidence levels. This visualization enables a thorough evaluation of the models’ performance across different confidence levels.

Figure 9a presents a comparison of precision between RDD-YOLOv5 and the ablation models. The curve demonstrates that RDD-YOLOv5 outperforms YOLOv5, particularly in medium and high confidence levels. This highlights the ability of RDD-YOLOv5 under challenging conditions. At a high confidence level, the precision curve exhibits an early rise compared to YOLOv5, indicating that the optimized model is able to detect more road defects in a precise way. Notably, the k-means++ algorithm seems to contribute more to the model in low confidence. This is because the K-means++ algorithm demonstrates distinct advantages when dealing with approximately similar rectangular bounding boxes. Three ablation models display different degrees of performance superiority compared to YOLOv5, which demonstrates the validity of ablation models.

Figure 9b portrays the recall curve under the confidence level. The recall rate reflects the ability to recall true positive samples (TP) and reduce false negative rate (FN). A high recall at low confidence shows that the model is effective at reducing false negative and miss rates, resulting in a higher proportion of ground truth being captured. In contrast to the precision curve, RDD-YOLOv5 shows a notable increase in recall at low confidence, which is better than YOLOv5. This improvement is particularly advantageous for road defect detection as it enables the acquisition of more comprehensive feedback regarding inspection information. Likewise, every ablation model collectively contributes to the improvement in the recall rate.

Figure 9c is the combination of (a) and (b), adopting a holistic perspective that considers the interrelationship between precision and recall. Ideally, authors aspire for the model to maintain high precision while achieving a high recall rate. Multiple improvement strategies exhibit a notable cumulative effect on RDD-YOLOv5′s recognition capability, reaching an optimal level of model performance, as shown in the figure that the RDD-YOLOv5 curve is considerably higher than ablation models’ curves. It is demonstrated that RDD-YOLOv5 has superior performance over YOLOv5 at equivalent levels of precision or recall.

Figure 9d calculates the F1 score of multiple models, which is a comprehensive reflection of both recall and precision; the F1 score equation is depicted as Equation (24). It can be visualized that the EVC Module attains a higher level of F1 score across broader confidence intervals, while RDD-YOLOv5 decreases this trend, showing a moderate performance in the whole confidence. There is a notable improvement in both high and low confidence levels. This further substantiates the effectiveness of the three model improvement strategies.

In summary, the proposed YOLOv5 improvement strategies in this study demonstrated a strong performance in the ablation experiments. Particularly, the RDD-YOLOv5 model exhibited excellent results in the precision–recall curve in Figure 9c, indicating significant synergistic effects of multiple modules in the model. Hence, the introduction of RDD-YOLOv5 in this research is reasonable.

### 5.6. Comparison Experiments

To further ascertain the superiority of RDD-YOLOv5 in the context of road defect detection, experiments were conducted by training and evaluating in the same research dataset. Comparison experiments involve mainstream object detection algorithms such as SSD [52], Fast R-CNN [54], Mask R-CNN [23,65], YOLOv3 [68], YOLOv4 [69], YOLOv5, and YOLOv7 [70]. It is noteworthy that Mask-RCNN has been applied in the Global Road Damage Detection competition [23]. Results are shown in Table 7. Compared to YOLOv5 and the traditional detectors predating YOLOv5, RDD-YOLOv5 exhibited remarkable outcomes, surpassing the original model in every aspect, including the F1 score, mAP@0.5, and mAP@.5:.95. In comparison to RDD-YOLOv5 and more advanced detectors like YOLOv7, RDD-YOLOv5 is slightly less impressive. Taken as a whole, RDD-YOLOv5 achieves better results with a smaller model volume, considering the lightness and ease of deployment, demonstrating a stronger overall performance.

To demonstrate the adaptability and superior performance of the proposed model on different datasets, this experiment applied the CrackTree200 [6], CrackForest Dataset [71], and ChinaDrone Dataset from RDD2022 [72] to evaluate and compare RDD-YOLOv5 with YOLOv4 and YOLOv5 in 400 training epochs and Mask-RCNN in 12 rounds. The results are shown in Table 8. Among the four datasets used in the RDD-YOLOv5 model, the Road Defect Dataset exhibits the best performance, achieving an accuracy of over 90%. The performance on the other datasets is relatively average. It is worth noting that the labels for CrackTree200 and CrackForest Dataset were created by the authors of this paper, which introduces a significant level of subjective judgment. Therefore, they cannot serve as the ultimate benchmark for evaluation. However, in comparison to the benchmark dataset ChinaDrone in GRDD2022, the Road Defect Dataset shows outstanding performance. Thus, the Road Defect Dataset initially meets practical application standards. Although the dataset may not cover a comprehensive range of categories, it adequately fulfills basic road detection requirements. Furthermore, when comparing the same dataset across multiple models, RDD-YOLOv5 demonstrates better recognition capabilities. These results strongly attest to the advantages of this research.

## 6. Discussion

Aiming at the challenges conventional object detection algorithms face in ensuring the detection of road defects in complex traffic scenarios and the need for lightweight deployment via UAV inspection, this research submits a novel approach. A specialized dataset named the Road Defect Dataset was collected, and an object detection algorithm was devised for road defect detection.

This research builds upon the YOLOv5 model, employing three strategies. Firstly, SW Block with shift-window based multi-head self-attention mechanism is embedded as an efficient extraction module. Secondly, it integrates the explicit vision center into a multi-scale feature fusion network to facilitate feature fusion and precise object localization. Thirdly, SiLU activation is replaced by the GELU at the cost of the sacrifice of training speed. To address the irregularity in image label aspects compared to standard datasets, an enhanced K-means algorithm is employed to compute the optimal initial anchor boxes. In the experimental part, this research introduces various techniques, such as label smoothing and models for the flight altitude and speed of UAVs, which contribute to the comprehensive process of the proposed approach.

Comparison and ablation experiments are conducted to validate the effectiveness of the project detection algorithm and the dataset. The ablation studies demonstrated that the algorithmic improvements proposed in this research have a cumulative contribution to enhancing the algorithm’s performance. Specifically, K-means++, SW Block, EVC Block, and CBG provide improvement effects of 1.71%, 0.93%, 1.99%, and 0.74%, respectively. The comparison experiments consist of two parts. One is to demonstrate the superiority of RDD-YOLOv5 by comparing it with several mainstream models (Mask-RCNN, Faster R-CNN, SSD, YOLOv3, YOLOv4, YOLOv5, etc.), and the results show that the proposed model achieves an mAP of 91.5%. Compared to the original model, the mean average precision (mAP) was improved by 2.5% with a model volume of 24 MB, which shows a relatively lower model size for lightweight road detection based on UAV inspection. While the mAP may not be as high as that of YOLOv7, it offers the advantage of lightweight detection suitable for real-time deployment on drones. The second part compared the Road Defect Dataset (RDD) with several open-source road datasets, including CrackTree200, CrackForest Dataset, and China Drone in GRDD2022. Multiple experiments demonstrated that these open-source datasets generally suffer from accuracy issues and are challenging to use for practical road detection. RDD-YOLOv5 consistently exhibited excellent performance across multiple datasets, showing improvements ranging from 1% to 4% in detection accuracy.

## 7. Conclusions

This paper presents a comprehensive approach to road defect detection, addressing various challenges through the construction of a diverse dataset, data augmentation methods, and improvements to the YOLOv5 algorithm. The dataset’s richness and specificity make it a valuable resource for drone-based road detection tasks, while the proposed data augmentation techniques tailor the dataset for unmanned aerial vehicle (UAV) road defect images. In the task of road defect detection during unmanned aerial vehicle (UAV) inspections, RDD-YOLOv5 achieves an impressive accuracy of 91.5% while successfully meeting the lightweight model target with a reduced size of only 24 MB. By optimizing the network, it demonstrated the ability to accurately predict the specific types and locations of road defects, ensuring practical and efficient detection. The establishment of UAV flight height and speed equations further enhances the quality and effectiveness of the detection process without increasing computational power, making the algorithm more suitable for real-world applications. This research provides an integrated and complete road detection method, serving as a technological reference for drone-based road inspections.

While the proposed method achieved favorable results in practical road defect detection, there are drawbacks and opportunities for further improvement. (1) The proposed model faces the challenge of a relatively large number of parameters, which poses certain requirements on the devices it runs on. So, it is necessary to reduce the model volume of RDD-YOLOv5 through pruning and optimization efforts to make it more lightweight. (2) The inherent drawback of drone inspection is its limited battery life, making it challenging to undertake long-distance continuous road inspection tasks. Drones are more suitable for short-distance inspections in complex urban scenarios. (3) The dataset contains fewer types of road defects, which may not be sufficient to address the diverse range of road surface inspection tasks in urban environments. For example, the Road Defect Dataset does not include certain types, such as asphalt oil spills and loose road surfaces, which will pose certain challenges for the detection task. Additionally, expanding the dataset to cover a broader range of defect types will increase its universality and applicability.

However, future road defect detection tasks will demand even higher precision in localizing defects, considering complex traffic environments and potential occlusions by vehicles. Addressing challenges in detecting multiple and small objects in complex scenarios will be critical in advancing the field. Exploring the application of object segmentation algorithms to calculate the sizes of road defects and evaluate road damage extent will contribute to the algorithm’s practicality and accuracy.

## Figures and Tables

**Figure 1 sensors-23-08241-f001:**
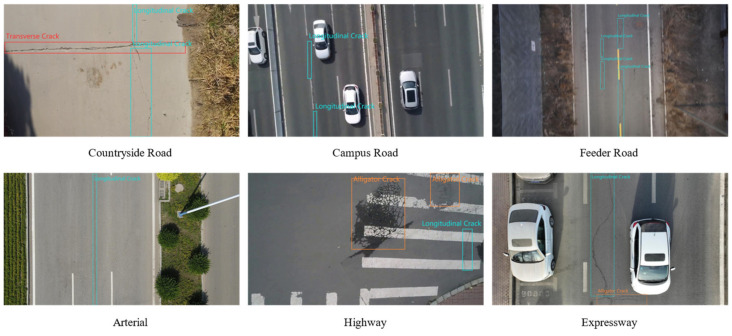
The Road Defect Dataset contains six scenarios: countryside road, campus road, feeder road, arterial, highway, and expressway. The variety and complexity of the scenarios improve the robustness of the model.

**Figure 2 sensors-23-08241-f002:**
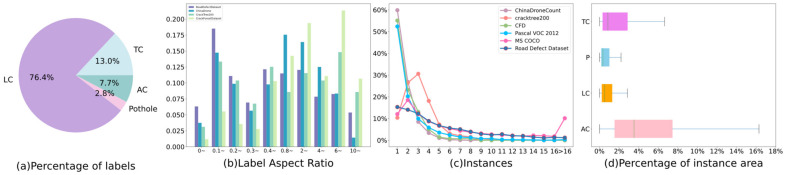
Statistical patterns of Road Defect Dataset. (**a**) Pie chart of the percentage of the four defect classes. (**b**) Histogram of the aspect ratio distribution pattern of the instances, including Road Defect Dataset, China Drone, CrackTree200, and CrackForest Dataset. (**c**) Line chart of the distribution of individual image instances across different datasets. (**d**) Box plot of the proportion of road defect categories within images.

**Figure 3 sensors-23-08241-f003:**
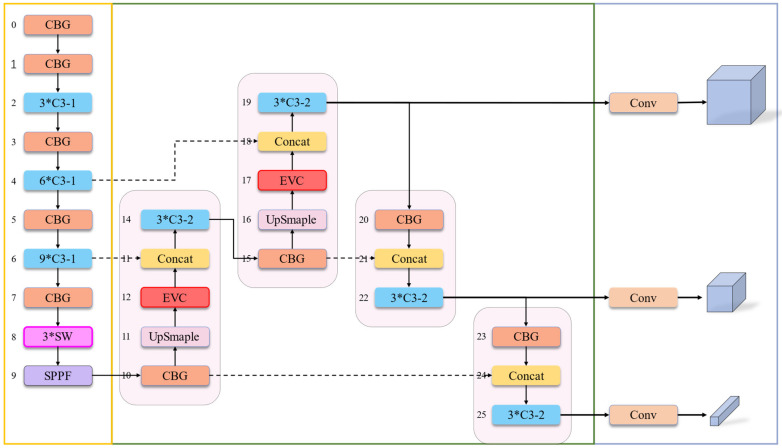
Structure of RDD-YOLOv5. SW Block is the swim transformer block, which contains the shifted windows multi-head self-attention. EVC Module is the explicit vision center, focusing on capturing global and local information. CBG, as a basic module, is the combination of the convolution layer, batch normalization, and GELU activation.

**Figure 4 sensors-23-08241-f004:**
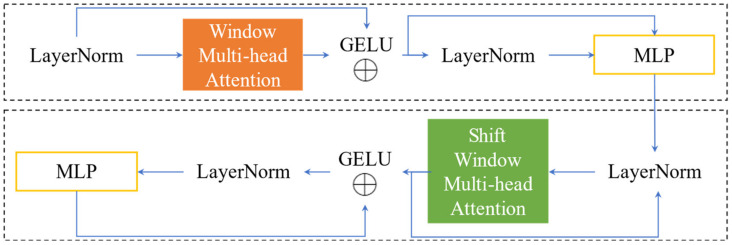
Structure of SW Block in RDD-YOLOv5. SW Block consists of window and shift window multi-head self-attention. This figure simplifies the process of applying residual connections to each module.

**Figure 5 sensors-23-08241-f005:**
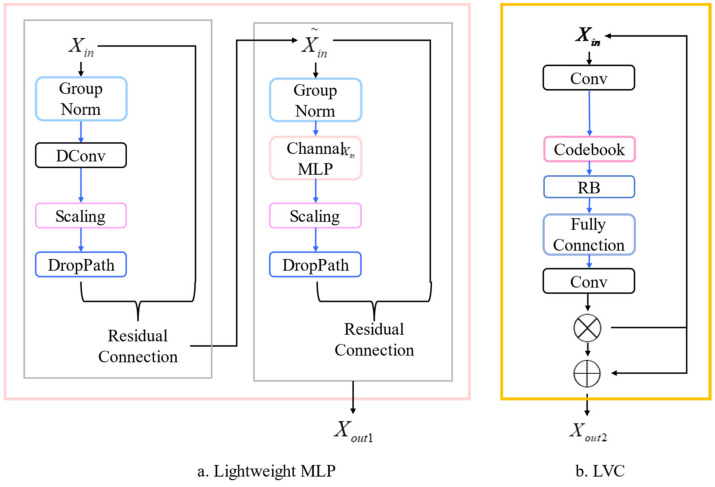
The structure of two modules of EVC Module. (**a**) The structure of Lightweight MLP. $X_{out1}$ represents the first output from lightweight MLP. (**b**) The structure of LVC. ⊕ represents the channel-wise addition. ⊗ represents the channel-wise multiplication. $X_{out2}$ is the output of LVC.

**Figure 6 sensors-23-08241-f006:**
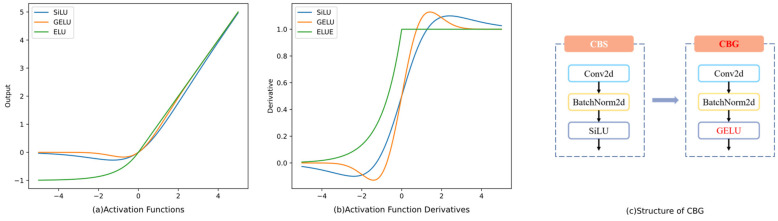
Comparison of SiLU, ELU, and GELU activations between the original function and derivative function and the structure of CBG. (**a**) Comparison between SiLU, ELU, and GELU activations. (**b**) Comparison of SiLU, ELU, and GELU activations in the derivative function. (**c**) Structure of CBG (Convolution + BatchNormalization + GELU).

**Figure 7 sensors-23-08241-f007:**
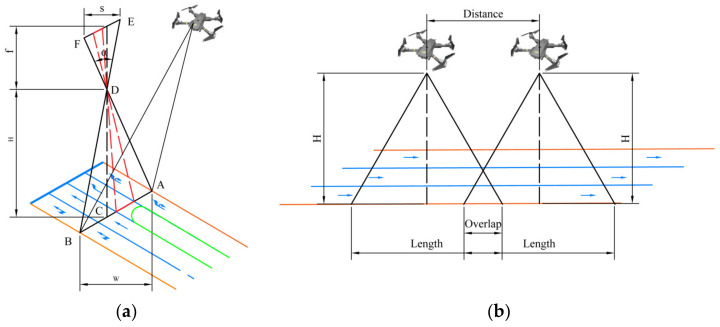
The diagram of the UAV flight altitude and flight speed. The blue and orange lines represent the road lines. The green lines represent the road green belt. (**a**) Camera shooting height diagram. (**b**) UAV continuous capturing and image overlap schematic.

**Figure 8 sensors-23-08241-f008:**
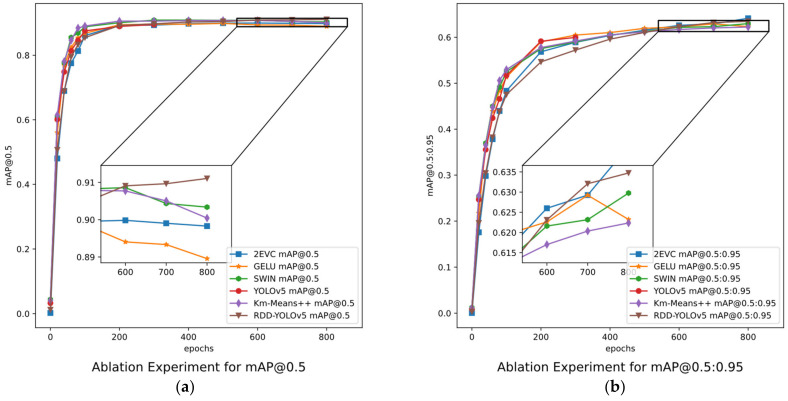
The performances of strategies in YOLOv5s in ablation experiments. (**a**) is the evaluation of mAP@0.5 and (**b**) is mAP@0.5:0.95. They represent the ability of individual modules under different metrics.

**Figure 9 sensors-23-08241-f009:**
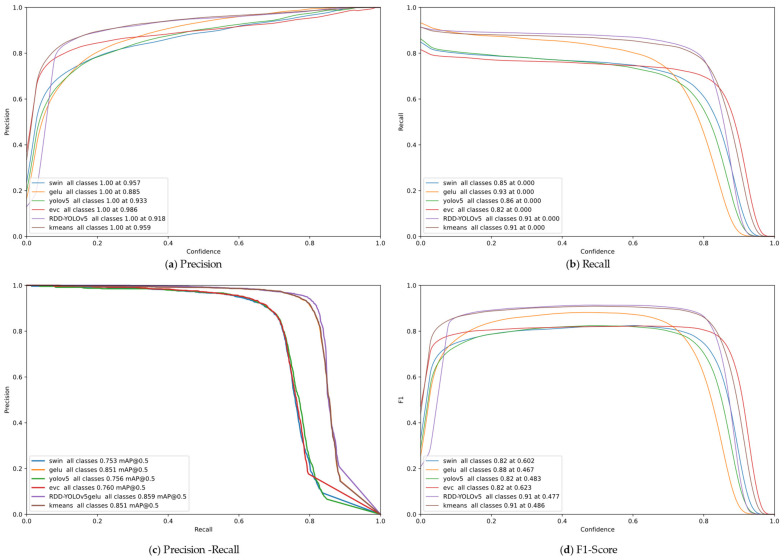
Evaluation metrics for different modules in the ablation experiments at various confidence levels. (**a**) displays the precision. (**b**) displays the recall. (**c**) represents the relationship between precision and recall. (**d**) represents the ablation models of F1 score (2 ∗ P ∗ R/(P + R)).

**Table 1 sensors-23-08241-t001:** Parameter for each layer of RDD-YOLOv5. The “Params” represents the amount of parameter calculations for the block. The “Arguments” include channel, kernel size, stride and padding.

From	n	Params	Module	Arguments
−1	1	3520	CBG	64, 6, 2, 2
−1	1	18,560	CBG	128, 64, 3, 2
−1	1	18,816	C3	128
−1	1	73,984	CBG	256, 3, 2
−1	2	115,712	C3	256
−1	1	295,424	CBG	512, 3, 2
−1	3	625,152	C3	512
−1	1	1,180,672	CBG	1024, 3, 2
−1	2	2,109,456	SW Block	1024
−1	1	656,896	SPPF	1024, 5
−1	1	131,584	CBG	512, 1, 1
−1	1	0	Upsample	None
−1	1	4,287,680	EVC Block	256, 256
[−1, 6]	1	0	Concat	1
−1	1	361,984	C3	512, 256, 1, False
−1	1	33,024	CBG	256, 128, 1, 1
−1	1	0	Upsample	None
−1	1	1,077,952	EVC Block	128, 128
[−1, 4]	1	0	Concat	1
−1	1	90,880	C3	256, 128, 1, False
−1	1	147,712	CBG	128, 128, 3, 2
[−1, 15]	1	0	Concat	1
−1	1	296,448	C3	256, 256, 1, False
−1	1	590,336	CBG	256, 256, 3, 2
[−1, 10]	1	0	Concat	1
−1	1	1,182,720	C3	512, 512, 1, False
[19, 22, 25]	1	16,182	Detect	anchors

**Table 2 sensors-23-08241-t002:** Flight altitude clearance requirements. This table includes the heights of the vast majority of obstacles.

Infrastructure	Traffic Signs	Traffic Lights	Street Lights	Roadside Trees	Utility Poles
Altitude (m)	4.5	3~7	6~12	<15	12~17

**Table 3 sensors-23-08241-t003:** GSD parameters for three common cameras; H refers to the flight height.

Camera Lens	24 mm	35 mm	50 mm
GSD	H/55	H/80	H/114

**Table 4 sensors-23-08241-t004:** Flight setup for 24 mm focal length camera. There are two main classes of roads in the dataset: low-grade roads and high-grade roads. Due to variations in the average width and obstacle height between these road types, different flight altitudes and speeds are required for effective inspection.

Road Classes	Sensor Size (mm)	Focal Length (mm)	Flight Altitude (m)	Flights Speed (m/s)	t (s)
Low-grade roads	17.3 × 13	24	5~8	1.2	2
High-grade roads	17.3 × 13	24	15~20	1.2	2

**Table 5 sensors-23-08241-t005:** The anchor size parameters table. The results are clustered by K_means algorithm.

Object Scale	Anchor Size
Large	19,27, 59,14, 17,64
Medium	39,64, 24,122, 132,23
Small	32,280, 361,44, 54,316

**Table 6 sensors-23-08241-t006:** Experimental results of ablation experiments with YOLOv5s under three strategies. This table records the performance of k-means++, SW Block, EVC Block, and GELU in the model, respectively. Metrics include precision, recall, mAP@0.5, and mAP@.5:.95.

Model	K-Means++	SW	EVC	GELU	Precision	Recall	mAP@0.5	mAP@.5:.95
YOLOv5	🗴	🗴	🗴	🗴	94.61	85.88	89.17	60.16
	✓	🗴	🗴	🗴	96.15	87.71	90.89	62.37
	🗴	✓	🗴	🗴	96.47	87.48	90.10	60.22
	🗴	🗴	✓	🗴	95.93	87.19	91.16	65.66
	🗴	🗴	🗴	✓	95.81	86.26	89.91	62.93
	✓	✓	✓	✓	96.32	88.85	91.68	64.12

**Table 7 sensors-23-08241-t007:** Experimental results of comparing RDD-YOLOv5 with other algorithms. The comparison experiments were conducted by using the Road Defect Dataset to demonstrate the superiority of RDD-YOLOv5 over the current mainstream object detection algorithms.

Model	mAP@0.5 (%)	mAP@0.5:v0.95 (%)	F1 Score (%)	Volume (MB)
Mask R-CNN	\	\	51.50	311.9
Faster R-CNN	63.46	\	\	90.7
SSD	57.12	\	\	92.1
YOLOv3-tiny	79.74	40.78	79.99	16.6
YOLOv4-cfp-tiny	68.45	34.33	60.00	45.1
YOLOv5	89.17	60.16	90.48	14.3
YOLOv7	91.42	65.78	92.68	73.1
RDD-YOLOv5	**91.68**	**64.12**	**92.43**	**24.0**

**Table 8 sensors-23-08241-t008:** Experimental results of different datasets in RDD-YOLOv5 model and other mainstream object detection algorithms. CrackTree200, CrackForest Dataset, and China Drone Dataset in Global Road Damage Dataset 2022(GRDD2022) are applied on the Mask YOLOv3, YOLOv4, YOLOv5, and RDD-YOLOv5 to test the universality of RDD-YOLOv5 and Road Defect Dataset.

Dataset	Model	mAP@0.5 (%)	mAP@0.5:0.95 (%)	Volume (MB)
CrackTree200	Mask R-CNN	\	59.10	311.7
	YOLOv4-cfp-tiny	64.76	37.38	22.4
	YOLOv5	61.76	34.42	13.7
	**RDD-YOLOv5**	**65.32**	**34.59**	**25.8**
CrackForest Dataset	Mask R-CNN	\	59.10	311.8
	YOLOv4-cfp-tiny	62.00	20.17	22.4
	YOLOv5	58.78	37.01	13.7
	**RDD-YOLOv5**	**63.54**	**37.94**	**25.8**
ChinaDrone	Mask R-CNN	\	38.00	311.8
	YOLOv4-cfp-tiny	63.56	35.42	22.4
	YOLOv5	60.45	38.39	13.7
	**RDD-YOLOv5**	**62.76**	**40.35**	**25.9**

## Data Availability

https://github.com/keaidesansan/Roadcrack_Dataset_2517.git (accessed on 26 September 2023).

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
