# Peer review of "RDD-YOLOv5: Road Defect Detection Algorithm with Self-Attention Based on Unmanned Aerial Vehicle Inspection"

_sensors, 2023, doi:10.3390/s23198241_

Round 1
Reviewer 1 Report
The article titled "RDD-YOLOv5: Road Defect Detection Algorithm Based on 2 YOLOv5" addresses the important problem of road defect detection.
The main contributions of this research, as the authors state, are:
* A Road defect dataset is built. It includes common category of road dataset and covers multiple road background and traffic conditions with precise manual annotations. To ensure its validity and universality, a few tricks of data augmentations are implemented on the dataset, which is suitable for road defect images based on UAV perspective.
* An improved YOLOv5 algorithm named RDD-YOLOv5 (Road Defect Detection YOLOv5) is proposed. Considering the complexity and diversity of road defects, bottleneckCSP(C3) is replaced with a self-attentive mechanism model called SW Module, and spatial explicit vision center called EVC Module is added to the neck network to capture long-range dependencies and aggregate local critical regions. Finally, the activation is replaced with GELU.
* In the experiment, an UAV flight platform is established including accurate mathematical model for flight altitude and flight speed to improve the efficiency and image quality of UAVs in collecting road defects images. A few experimental tricks are provided to further enhance the performance. In this case, the anchors were recalculated to improve the positioning accuracy for the abnormally shaped bounding boxes included in the dataset developed in this study. Additionally, label smoothing is apllied to mitigate the impact of manual annotations errors in training, contributing to enhancing precision.
A few suggestions/questions for the authors:
- The statement about data availability should be provided at the end of the article, that the dataset is available. The provided link to the dataset "https://github.com/arandinglv/RoadDefectDataset" leads to an empty page. Authors are encouraged to make the assembled road defect dataset available publicly through well-known scientific data-sharing services.
- The proposed road defect detection algorithm could be compared against other similar solutions using the same dataset. There are at least 12 projects with code in the database of Papers with Code, Road Damage Detection section (https://paperswithcode.com/task/road-damage-detection). The solutions also could be compared using other datasets - the authors of the current article provide alternative datasets.
- The captions of Tables 4, 5; Figures 1, 3, 4, 8, 9 could be expanded to be more comprehensive and descriptive enough to be understood without having to refer to the main text.
- The annotations provided in Figure 1 are not clearly visible.
- Why did the authors detect the road defects as bounding boxes but not as semantically segmented image (labeled image pixels) or rotated bounding boxes? Regular bounding boxes would not label efficiently the cracks that are not aligned vertically or horizontally in the images.
Author Response
"Please see the attachment.

Reviewer 2 Report
- Please give layers and parameters of the proposed model as a table form.
-
Please give the limitations of the work.
-
There is no discussion about your results through the paper.
-
Please use more dataset such as Crack Tree-200, CFD, and CC.
-
Please use more performance metrics such as Structural SIMilarity (SSIM), Peak Signal to Noise Ratio (PSNR), Inception Score (IS), and Frechet Inception Distance (FID).
-
Please cite following papers:
-CFC-GAN: Forecasting Road Surface Crack Using Forecasted Crack Generative Adversarial Network
-AugMoCrack: Augmented morphological attentionnetwork for weakly supervised crack detection
-Feature pyramid network with self-guided attention refinement module for crack segmentation
-Estimation of Road Surface Type from Brake Pressure Pulses of ABS
-Guest Editorial: Special issue on big data in transportation
Moderate editing of English language required
Reviewer 3 Report
The manuscript titled "Road Crack Detection YOLOv5 (RDD-YOLOv5)" introduces a novel deep learning algorithm aimed at addressing the challenging task of road defect detection, marking a significant advancement in the field of road maintenance and safety. RDD-YOLOv5 combines a transformer structure, an explicit vision center, and Gaussian Error Linear Units to effectively tackle the complexities of road crack detection. The experimental results further validate the algorithm's superiority, positioning it as a valuable asset for road maintenance projects. Overall, RDD-YOLOv5 represents a noteworthy stride toward safer and more efficient road maintenance practices. Minor revisions are required to enhance the manuscript, and several key review points are outlined below:
(1) The manuscript contains substantial text repetition, which can hinder readability and increase time consumption. Streamlining the content is advisable to address this issue.
(2) The conclusion section lacks a comprehensive summary of the research papers and is deficient in presenting data related to the results. A more detailed and data-driven conclusion would strengthen the manuscript's overall impact.
(3) It is important to carefully review the manuscript for typographical mistakes to ensure clarity and professionalism in the final document.
English language quality is satisfactory.
Reviewer 4 Report
The following major revisions are proposed:
- The title needs to be a little more descriptive, as it leaves the topic too generic.
- Include more relevant references in the introduction and make it a little longer. Include a slightly more recent reference.
- The method followed should be explained in more detail. In particular the general description of the YOLOv5 model.
- A specific section on results should be established, where the results obtained are explained and detailed.
- More conclusions should be drawn, more clearly explained, based on the results obtained.
Round 2
Reviewer 2 Report
All my suggestions/comments have been addressed by the authors.
Reviewer 4 Report
The article is now proposed for publication after the suggested indications have been adequately addressed.